# Placental Malfunction, Fetal Survival and Development Caused by Sow Metabolic Disorder: The Impact of Maternal Oxidative Stress

**DOI:** 10.3390/antiox12020360

**Published:** 2023-02-02

**Authors:** Xizi Yang, Ruizhi Hu, Mingkun Shi, Long Wang, Jiahao Yan, Jiatai Gong, Qianjin Zhang, Jianhua He, Shusong Wu

**Affiliations:** Hunan Collaborative Innovation Center for Utilization of Botanical Functional Ingredients, College of Animal Science and Technology, Hunan Agricultural University, Changsha 410128, China

**Keywords:** sow, oxidative stress, pregnancy, maternal metabolism, placenta, fetal development

## Abstract

The energy and metabolic state of sows will alter considerably over different phases of gestation. Maternal metabolism increases dramatically, particularly in late pregnancy. This is accompanied by the development of an increase in oxidative stress, which has a considerable negative effect on the maternal and the placenta. As the only link between the maternal and the fetus, the placenta is critical for the maternal to deliver nutrients to the fetus and for the fetus’ survival and development. This review aimed to clarify the changes in energy and metabolism in sows during different pregnancy periods, as well as the impact of maternal oxidative stress on the placenta, which affects the fetus’ survival and development.

## 1. Introduction

Maintaining sow performance has become critical in the modern pig industry for meeting output targets. The number of piglets weaned per sow per year (PSY) is an important metric of how efficiently pig farms run and how well sows reproduce [1].

Genetic selection and early weaning have made it possible for hyper-prolific sows to have PSYs greater than 31 [2], and this may cause a greater metabolic burden for sows, and possibly a shorter productive life. Simultaneously, the larger the litter size, the lower the uterine blood flow per fetus and piglet birth weight, and the more variation in piglet birth weight within the litter, which raises management expenses in modern all-in-all-out swine production systems [3,4,5]. The increased metabolic demands on sows have a number of detrimental consequences, including oxidative stress and decreased reproductive efficiency. Therefore, the key to increasing the efficiency of pig production is to reduce the metabolic burden of sows while maintaining the reproductive performance of high-yield sows. Pregnancy is a dynamic and finely coordinated process in which maternal metabolism plays a critical role in fetal survival and growth [6]. Maternal metabolic status varies throughout pregnancy, with major alterations in lipid and glucose metabolisms that facilitate nutrition supply to the developing fetus and have a significant impact on reproductive function [7,8]. The maternal endocrine state during gestation also undergoes drastic changes, which has an important impact on the maintenance of pregnancy and fetal development. Maternal metabolism during early pregnancy is predominantly anabolic, with catabolic circumstances common in late pregnancy [9]. Metabolic diseases can impede fetal and neonatal growth and increase morbidity and death at any stage of pregnancy [10]. The placenta serves as the vital link between the fetal and the maternal circulations, sustaining pregnancy and embryonic development and growth by facilitating the transfer of nutrients and waste, contributing to immunological response, and acting as an endocrine organ [11,12]. There is evidence that maternal factors influence the morphology and function of the placenta [13]. This paper discusses changes in energy supply and metabolism in sows during pregnancy, as well as the consequences of oxidative stress-induced placental malfunction on fetal survival and development.

## 2. Maternal Energy and Metabolism Changes during Pregnancy

### 2.1. Early Pregnancy

The first 30 days of pregnancy in sows is the phase of early pregnancy, which is the critical stage of embryo development and implantation. Sows ovulate roughly 20–30 follicles, and with a fertilization rate close to 90%, the number of early embryos in sows can theoretically reach 18–27; however, due to different variables, the litter size only reaches half of the embryos [14]. As a result, the survival rate of early embryos has a significant impact on sow litter performance. It is generally agreed that the fully functional corpus luteum (CL) is essential for the establishment of pregnancy and early embryonic development in all mammals [15]. Porcine CL grows rapidly and gains maximum size between day 10 and 12 after ovulation, weighing approximately 7 g in gilts and 13 g in multiparous sows [16]. Luteinizing hormone (LH) has a role in the maintenance of CL [17]. A study found that suboptimal progesterone concentrations as a result of poor pituitary LH support during pregnancy days 12–18 interfere with proper embryo growth and their ability to create signals necessary for pregnancy recognition [18]. CL provides uninterrupted synthesis and the release of progesterone to stimulate the proliferation of endometrial cells and stabilize the uterus for embryo implantation and pregnancy maintenance [19]. Embryos signal their presence by secreting estrogens, which triggers the pregnancy recognition process [16]. Additionally, as estrogen and progesterone levels rise during the early pregnancy stage, tissues become more sensitive to insulin [20]. Moreover, it is necessary for the remodeling of the endometrial lining in order to assist implantation and the delivery of nutrients to the embryos, which improves embryo survival [16].

Sows exhibit overall anabolic features during pregnancy, particularly in the early stages when nutrients tend to accumulate in the maternal. Maternal metabolism is important in the success of early embryonic development because it creates an optimal condition for the establishment and maintenance of pregnancy [21]. In early pregnancy, changes in basal and postprandial glucose metabolism occur gradually, energy reserves build up in the maternal, and insulin sensitivity increases slightly to accommodate the growing nutritional needs of both the maternal and the fetus [22]. Early pregnancy, in particular, is linked to adipocyte hypertrophy, enhanced lipogenesis, and lipid storage, as well as improved insulin sensitivity of white adipose tissue in the maternal [23]. The first 24–48 h after mating are critical for embryo implantation [24], it is necessary to limit the feed intake to avoid an increase in progesterone catabolism produced by increased hepatic blood flow, which reduces embryo survival rate [25]. Glucose has been demonstrated to affect early embryo development and quality [26]. One study found that cultured porcine early embryos with 7 mM of glucose for 48 h generated endoplasmic reticulum stress and oxidative stress, which further impacted embryo quality and development [27]. However, Diego et al. [28] discovered through meta-analysis that feeding diets with more energy than needed for body maintenance have no negative effect on embryonic survival in the majority of trials. Moreover, within 34 days of insemination, the energy intake of sow gilts (Yorkshire x Landrace) as high as 54 MJ/day showed no deleterious influence on embryo development and improved the pregnancy rate [29]. Increasing sows’ energy intake levels after 48 h of pregnancy aids in the recovery of early backfat. Moreover, from day 3 to day 28 of pregnancy, primiparous sows (Landrace × Large White) with higher energy intake (3.75 kg/d of a diet with 2.18 Mcal NE/kg) had a higher number of live-born piglets than those with lower energy intake (3.125 kg/d of a diet with 2.18 Mcal NE/kg) [30]. Therefore, the development of the embryo benefits from the increase in sows’ energy intake during the early stages of pregnancy.

### 2.2. The Second Trimester of Pregnancy

The direction of nutrient deposition starts to change during the second trimester of pregnancy to support the maternal body reserve restoration. A relatively low energy intake during the second trimester of pregnancy should be preserved to maintain the sow’s optimal body condition and prevent excessive back fat deposition in sows. A greater rate of stillbirths was observed when gestating gilts (Yorkshire x Landrace) were fed a 2.5 kg/d basal diet containing 3265 kcal of ME/kg as compared to feedings of a 2.0 kg/d diet [31]. Moreover, a study found that feeding sows (Landrace × Large White) an extra 2 kg/d of feed during mid-gestation reduced milk production capability and suckling piglet survival rate [32]. The maternal blood glucose rises in the second trimester of pregnancy [33], and the recovery to normal levels after meals is slow, indicating that insulin sensitivity is gradually impaired. The higher maternal glucose levels in the second trimester of gestation were associated with an increased risk of perinatal complications [34]. According to a human study, the normal maternal blood glucose level in the second trimester of pregnancy was 6.2 mmol/L, slightly higher than 5.7 mmol/L in the early pregnancy; however, the maternal blood glucose level in the second trimester of gestation rises to 8.3 mmol/L, which may be associated with adverse pregnancy outcomes [35]. Sows’ energy intake should be adjusted appropriately throughout this period of pregnancy to preserve the optimal body condition of sows.

### 2.3. Late Pregnancy

Most fetal weight gain occurs in late pregnancy. Sow metabolism increases dramatically in late pregnancy, maternal insulin sensitivity decreases further, and maternal catabolism satisfies the needs of rapid fetal growth. In late pregnancy, maternal fasting blood glucose levels fall [36]. However, the peak blood glucose level after a meal is high, and the rate of glucose clearance reduces as pregnancy progresses [37], indicating that insulin sensitivity declines. Sows in normal condition require more energy to maintain high blood glucose levels, which is conducive to the transmission of glucose to the fetus via the placenta and meets the fetus’s rapid growth in late pregnancy. Dietary energy intake during late gestation influenced the subsequent reproductive performance in multiparous sows [38]. A study has indicated that the piglet birth weight of sows (PIC 1050) with 6.75 Mcal NE/d energy intake in late pregnancy was 0.2–0.4 kg higher than that of sows with 4.5 Mcal NE/d [39]. Sows (Landrace x Yorkshire) fed 9600 kcal ME/d from day 85 of gestation till farrowing had higher body weight on day 110 of gestation and piglet weight than sows fed 8374 kcal ME/d [40]. However, excessive maternal energy intake in late pregnancy also thickened sows’ backfat, which raised the number of piglets born with low birth weight [41]. Furthermore, a study revealed that sows (Landrace × Large White) with higher feed intake (4.0 kg/d, 11.40 Mcal/d ME) had higher levels of reactive oxygen species (ROS) and malondialdehyde (MDA) in plasma at parturition compared to sows with lower feed intake (2.8 kg/d, 7.98 Mcal/d ME), which resulted in maternal oxidative stress [42].

The endocrine system also undergoes significant alterations in late pregnancy in order to support the fetus’ rapid growth and prepare for maternal delivery. Leptin is recognized as an important component linking metabolic status to reproduction, peaking between mid and late pregnancy in humans and mice, controlling peripherally the balance of energy reserves by facilitating pregnancy-specific endocrine responses [43]. The maternal follicle-stimulating hormone (FSH) works in tandem with LH to increase estrogen secretion. A study found that the levels of all three hormones in Bama mini pigs fluctuated continuously throughout the pregnancy period, peaking in the middle and late stages [44]. Saleri et al. [45] discovered that progesterone levels in sows (Large White x Landrace) declined in late pregnancy and were 7.38 ± 0.54 ng/mL at delivery. They also measured blood plasma prolactin levels, which revealed that prolactin levels increased two weeks before farrowing to reach a high at farrowing (67.83 ± 4.91 ng/mL), which is required for normal parturition, advantageous to mammary gland development, and prepares for postpartum lactation [45,46].

### 2.4. Progressive Oxidative Stress during Pregnancy of Sows

Table 1 summarizes some research findings on oxidative stress indicator levels and antioxidant capacity in sows during various phases of pregnancy. It has been discovered that progressive oxidative stress in sows occurs primarily in late pregnancy. Oxidative stress occurs when the amount of ROS produced exceeds the ability of antioxidants to neutralize it. Excess ROS causes oxidative damage to proteins, lipids, and DNA, and accelerates the pace of telomere shortening in prenatal tissues, and eventually impairs cell and organ functions [47,48]. The level of oxidative stress is determined by the balance between pro-oxidants and antioxidants.

As previously stated, the rapid development of the fetus, increase in energy intake of sows, and acceleration of maternal metabolism all culminate in increased ROS generation in late pregnancy. To ensure fetal growth and mammary development, increased rates of maternal digestion, absorption, and tissue mobilization result in an increase in ROS generation [50,53,54]. Furthermore, the antioxidant system of sows in late pregnancy declined, as did plasma antioxidant enzyme activity and total antioxidant capacity [54,55], potentially causing oxidative stress in the maternal. According to a study, the serum levels of ROS, DNA damage markers 8-hydroxy-deoxyguanosine (8-OHdG), and thiobarbituric acid reactive substances (TBARS) in sows were higher at 90 and 109 days of gestation than at 10 and 60 days [53], indicating that sows experienced increased systemic oxidative stress during late gestation. In late pregnancy, the function of the maternal non-enzymatic system also reduces. A study found that the plasma α-tocopherol and retinol concentrations of sows decreased from 7.14 and 1.10 mol/L (30 days of gestation) to 3.07 and 0.57 mol/L (110 days of gestation), respectively [54]. Sows are prone to being overweight in late pregnancy, which causes structural, metabolic, and functional changes in various tissues and organs, aggravating progressive oxidative stress [56], insulin resistance [57], and systemic inflammation [53] with high levels of proinflammatory cytokines, such as tumor necrosis factor-alpha (TNF-α), interleukins (IL-1, IL-6, and IL-8), and C-reactive protein (CRP) [58,59], leading to placental lipid toxicity [60]. Excessive back fat in sows during late pregnancy causes a lipid-toxic placental environment characterized by lipid accumulation, increased inflammation and oxidative stress, and insulin sensitivity reduction. These variables influence the expression of nutrition transport-related proteins such as the glucose transporter [61], fatty acid transporter [62], and amino acid transporter [63], which has a negative impact on nutrition transmission between the maternal and fetus, leading to undesirable consequences in pregnancy outcomes. Zhou et al. [41] collected data on 846 farrowing multiparous Yorkshire sows with parity ranging from 3 to 5, and found that the increased backfat thickness of sows at day 109 of gestation exhibited a convex quadratic relationship with litter size, litter weight, and average piglet birth weight. This demonstrates the connection between sows’ back fat thickness and litter performance. According to a study, high back fat sows (≥23 mm) had higher levels of triglycerides, malondialdehyde MDA, and pro-inflammatory factors in the placenta than normal back fat sows (17–22 mm). They also had reduced placental effectiveness on the 107th day of pregnancy, as well as smaller litters and piglets born with lower average weights than typical [64]. Therefore, the litter performance of sows is greatly influenced by the status and function of the placenta.

## 3. Placenta and Oxidative Stress

### 3.1. The Structural Characteristics of Placenta

The placenta is critical in transferring maternal nourishment to the fetus and regulating fetal growth and development [65]. The porcine placenta is diffuse, folded, without decidualization, non-invasive, and epitheliochorial, accomplished with interdigitations of the trophectoderm microvilli and surface uterine epithelium [66], which belongs to an epitheliochorial type, with its surface being attached to maternal endometrium, and six layers of tissues to separate the fetus from the maternal blood [67]. Endometrial epithelium, connective tissue, and vascular endothelium are found in the maternal part, while vascular endothelium, intermediate connective tissue, and trophoblast cells are found in the fetal part. At 30 gestation days, the pig placenta is fully developed [67], and a considerable number of microcapillaries begin to appear [66]. The formation and development of the placenta in early pregnancy are critical to the survival and healthy growth of the fetus (20–30% of fetal loss occurs during the early pregnancy) [68]. Placental growth occurs between days 20 and 70 of pregnancy, and by the second trimester (60 to 70 days), the placenta is close to its maximal size [69]. This is followed by rapid angiogenesis, which is a precursor to the rapid fetal growth that occurs in the latter stages of pregnancy [69]. At 90 days of gestation, the weight of the placenta does not grow [70]; however, the epithelial bilayer thins and capillaries indent the plane of each layer, reducing the distance between capillaries and between blood vessels and villi [71], facilitating nutrient exchange between maternal and fetal.

### 3.2. Mechanism of Oxidative Stress in Placenta

Throughout gestation, the placenta is continually evolving to accommodate the mounting demands of the fetus. In early pregnancy, the placenta and fetus exist in a hypoxic environment, the ROS produced by the placenta is low, and the O_2_ tension rises sharply at the end of early pregnancy when trophoblast invasion permits the occluded uterine spiral arteries to open [72]. As pregnancy progresses, the invasion of extra-villous trophoblast, the development of the placental vascular system, and the fetus’ metabolic demands increase, resulting in an increase in placental mitochondrial mass and mitochondrial electron chain enzyme activity, which leads to increased ROS production and placental oxidative stress [73]. Because of its high metabolic activity and frequent cell division, the placenta is very sensitive to oxidative stress [74]. The syncytiotrophoblast is especially vulnerable to oxidative stress because it is located on the surface of the villi, which is the first to encounter an increase in intervillous partial pressure of oxygen (PO_2_). On the other hand, syncytiotrophoblast has substantially lower quantities of antioxidant enzymes than other villous tissues [75,76,77]. Pregnancy is an oxidative stress condition, especially when the maternal is in the stage of rapid fetal growth. At the same time, the decline of maternal antioxidant capacity leads to an increase in ROS production, which aggravates the oxidative stress of the placenta [54,55].

Maternal insulin sensitivity gradually declines as pregnancy progresses, leading to insulin resistance, and increased triglyceride and cholesterol concentrations in late pregnancy produce lipid toxicity [78]. Lipid toxicity begins with lipid accumulation in non-adipose tissue. The placenta, like other organs such as the liver and muscle, is sensitive to obesity-related lipid accumulation [79,80]. Maternal obesity during pregnancy and insulin resistance lead to ectopic deposition of lipids to the placenta, causing placental lipid toxicity [41,81]. Placental lipid toxicity has been demonstrated to inhibit trophoblast invasion and influence placental development and transport, thus affecting fetal developmental pathways [82]. Placental lipid toxicity is also associated with augmented inflammation and oxidative stress. The results of RNA-sequencing performed on term placenta from obese or lean maternal revealed that maternal obesity increased placental lipid content and the expression of genes related to inflammation while decreasing antioxidant capacity [83]. Placental inflammation has been characterized by increased macrophage infiltration and increased cytokine production in the placenta. Studies have shown that lipid toxicity induces a pro-inflammatory response in placental cells that is regulated by JNK and EGR-1, increases the activation of inflammatory NF-κB signaling, increases pro-inflammatory cytokine levels such as IL-1, IL-6, and TNF-α, and decreases the total antioxidant capacity in the placenta [82,84,85]. The IKKα/NF-κB signaling pathway was also found to be implicated in MARK4-activated oxidative stress and mitochondrial dysfunction [86]. Placental oxidative stress increased placental mitochondrial activity and the production of ROS such as superoxide (O_2_•−), hydroxide (OH−•), and hydrogen peroxide (H_2_O_2_), which cause cellular damage and tissue malfunction and have a significant impact on placental function such as trophoblast proliferation and differentiation and vascular reactivity [87,88]. Mitochondria, a vital energy source for placental activity, are also the predominant producer of ROS. Excessive ROS can damage lipids, proteins, and nucleic acids within the mitochondria, resulting in alterations to mitochondrial structure and function. Furthermore, mitochondrial malfunction and superoxide overproduction could be part of a vicious cycle [89,90]. According to research, the consequences of increased oxidative stress on sow reproductive performance are mostly evident in reduced litter size and piglet survival, decreased ability for breastfeeding, and lower sow health status [91]. Increased oxidative stress is responsible for impaired milk production, reproductive performance, and finally, the longevity of sows [50]. During pregnancy, oxidative damage may predispose to embryonic resorption, limit fetal growth, and cause stillbirths [92].

### 3.3. Effects of Oxidative Stress on Placental Function

The exchange of nutrients, gases, and wastes between maternal and fetal circulation occurs through placental vessels, and a placenta with a high vascular density promotes the fetus’s growth and development. The porcine placenta belongs to an epitheliochorial type, with its surface being attached to the maternal endometrium, and six layers of tissues to separate the fetus from the maternal blood [67]. Increased levels of ROS have been shown to cause vascular endothelial cells to undergo autophagy, malfunction, and apoptosis, which may retard placental vasculature development [93]. The vascular endothelial growth factor receptor system (VEGF/VEGFR) plays a major role in the complex process of angiogenesis [94]. Oxidative stress induced by H_2_O_2_ prompted intracellular ROS generation and inhibited the tube formation and migration of porcine vascular endothelial cells (PVECs), as well as the expression of VEGF-A [95]. Placental insufficiency is more just inadequate blood flow; it may also result in diminished transplacental nutrition transfer capacity [96]. Passage across the placenta can occur via simple diffusion, pinocytosis, receptor-mediated uptake, and both the active and facilitative transporters, and substances required for fetal growth, such as glucose, fatty acids, and amino acids, are transported through specific uptake and transport mechanisms that can adapt to supply and demand [97]. Increased oxidative stress reduces GLUT1 expression and glucose transport in the pig placenta [98,99]. In addition, Cocl2-induced oxidative stress reduced mTOR signaling activity and LAT expression in HTR-8/SVneo human trophoblast cells [100]. Furthermore, reduced placental SNAT activity has been seen in pregnancies complicated by fetal growth limitation [101].

## 4. Effect of Oxidative Stress on Fetus

### 4.1. Fetal Development during Pregnancy

Early embryonic development in pigs, like in most mammals, consists primarily of different stages. The zygote, formed by fertilization, divides and develops into a blastocyst (embryonic day 0–5.5), the embryo attaches to the uterine wall (embryonic day 12), which is followed by the tissue development and organogenesis phase and embryo growth [102,103]. There are a large number of early embryos in the uterus during the post-implantation stage in pigs, in order to prevent intrauterine growth retardation (IUGR), a significant rise in embryo losses. The embryonic stage is followed by the fetal growth stage after organ differentiation and placental development. The fetal growth rate in sow early pregnancy is slow; during 35 days of gestation, the size of the embryo is roughly 3–4 cm and weighs about 5 g. The fetal length and weight in the second trimester of pregnancy (55 days) are approximately 11–12 cm and 90 g, respectively. At 90 days of gestation, the fetal length is about 25 cm and the weight is about 700 g [104]. Furthermore, in late pregnancy, the fetus enters the rapid development period, with fetal weight gain accounting for two-thirds of the birth weight. As a result, the first 30 days of sow pregnancy are crucial for embryo survival, whereas the latter 30 days of gestation are critical for increasing fetal birth weight. To satisfy the needs of self-maintenance, fetal growth, and development, the mother will undergo a series of severe changes in physiology, hormones, and metabolism [23]. The abnormality of metabolic alterations will have direct consequences for fetal health throughout the pregnancy, leading to fetal intrauterine growth retardation and low birth weight, as well as a long-term influence on offspring growth and development.

### 4.2. Effects of Maternal Oxidative Stress on Offspring in Sows

According to certain theories, oxidative stress is crucial to the pathophysiological process in offspring [105]. Increasing data suggest that unfavorable pregnancy outcomes are linked to maternal oxidative stress [106]. Zhao et al. [49] found that litter size and litter weight were negatively correlated with maternal oxidative stress indicators. Another study discovered that Large White sows with high backfat (21–25 mm) or low backfat (9–12 mm) at 110 days of gestation compared to medium backfat (13–20 mm) caused high levels of oxidative stress in the maternal, increased the number of stillborn piglets per litter, and decreased the number of piglets per litter, piglets born alive per litter, and litter birth weight [56]. Oxidative stress during gestation will impair sow health by increasing ROS production and buildup. Furthermore, ROS accumulation impairs fetal growth and development via the placenta, resulting in fetal death and intrauterine growth retardation (IUGR), and ultimately impairing reproductive performance [107]. Increased maternal MDA levels in plasma were linked to greater levels of IL-6 and IL-7 in the offspring, suggesting that maternal oxidative stress biomarkers may be associated with changes in cytokine concentrations in offspring [108]. Thus, oxidative stress may affect inflammatory pathways in progeny. IUGR is caused by utero-placental insufficiency; when placental tissue is unable to support appropriate nutrition and oxygen exchange, fetal organs grow slowly [109]. A suboptimal prenatal environment causes the placenta to experience cell stress and death, which compromises fetal growth, and causes metabolic organs to experience cellular stress and dysfunction, including oxidative stress and mitochondrial dysfunction [107]. The antioxidant defense mechanism of the IUGR fetus is impaired, aggravating oxidative stress [110]. Studies have revealed that oxidative stress in sows reduces litter sizes and the number of piglets born alive, which can be increased by 1.81 and 2.37, respectively, once maternal oxidative stress is relieved [111]; hence, decreasing oxidative stress in sows can lower IUGR in piglets [55]. The effect of oxidative stress on the fetus is shown in Figure 1.

## 5. Conclusions

In conclusion, the maternal, placenta, and fetus are inextricably linked and change dramatically as the pregnancy progresses. The maternal energy metabolism disruption, maternal progressive oxidative stress, and their consequences on placental function throughout pregnancy all have a significant impact on sow reproductive performance. The placenta plays a crucial role in the fetus’ survival and growth during pregnancy as it is a tissue that connects the maternal and the fetus and is susceptible to oxidative stress. The placenta serves as a link between the maternal and the fetus, and their interactions and mutual impacts jointly define mother and fetal health.

## Figures and Tables

**Figure 1 antioxidants-12-00360-f001:**
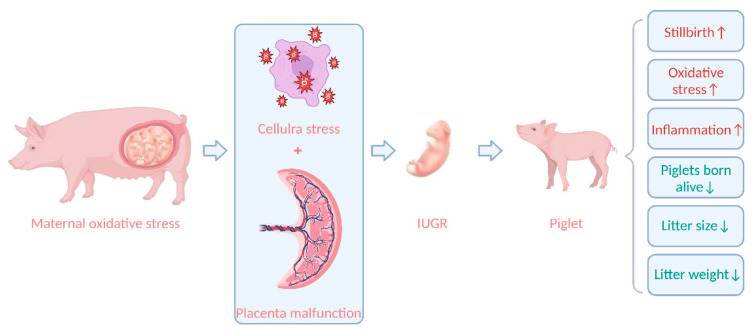
The effect of oxidative stress on the fetus. IUGR, intrauterine growth retardation. Figure 1 created with BioRender.com URL (https://app.biorender.com/).

**Table 1 antioxidants-12-00360-t001:** Oxidative stress markers of sows at different pregnancy stages.

Items			Early Pregnancy	The Second Trimester of Pregnancy	Late Pregnancy	References
Oxidative stress markers	Plasma	8-OHdG	0.71 ng/mL	1.00 ng/mL	1.07 ng/mL	Zhao et al. [49]
			-	0.72 ng/mL	0.69–0.95 ng/mL	Zhao et al. [50]
		MDA	-	5.50 μmol/L	5.79–5.67 μmol/L	Zhao et al. [50]
			-	6.35 nmol/mL	7.20 nmol/ml	Ostrenko et al. [51]
	Serum	ROS	About 1000 RLU	About 1000 RLU	About 3000 RLU	Tan et al. [52,53]
		TBARS	About 1.5 nmol/mL	About 1.5 nmol/mL	About 1 nmol/mL	Tan et al. [52]
		8-OHdG	About 21 ng/mL	About 35 ng/mL	About 40 ng/mL	Tan et al. [52]
Antioxidant capacity	Plasma	α-tocopherol	7.14 μmol/L	6.10 μmol/L	3.07 μmol/L	Berchieri-Ronchi et al. [54]
		Retinol	1.10 μmol/L	1.08 μmol/L	0.57 μmol/L	Berchieri-Ronchi et al. [54]
		GSH-Px	-	2376 units	2346 units	Ostrenko et al. [51]

8-OHdG, 8-hydroxy-deoxyguanosine, MDA, Malondialdehyde, ROS, reactive oxygen species, TBARS, thiobarbituric acid reactive substances, GSH-Px, Glutathione peroxidase.

## Data Availability

Not applicable.

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
