# Peer review of "Placental Malfunction, Fetal Survival and Development Caused by Sow Metabolic Disorder: The Impact of Maternal Oxidative Stress"

_antioxidants, 2023, doi:10.3390/antiox12020360_

Round 1
Reviewer 1 Report
The aim of the current article antioxidants-2160103 (Placental malfunction, fetal survival and development caused by sow metabolic disorder: the impact of maternal oxidative stress) deals with an interesting topic and fits with the scope of the journal.
The aim of this review study was to perform the changes in energy and metabolism during pregnancy in sows, and the effects of oxidative stress on fetal survival and development. However, the text of this manuscript should contain one -two tables, summarizing the results of previous studies or a diagram to present the effects of oxidative stress on fetuses. Moreover, the manuscript needs extensive revision for language and grammar, as well as in the design of the text.
In conclusion, I suggest a major revision based on the following major and minor comments.
Comments for the authors
Major comments
- the text of this manuscript should contain one -two tables, summarizing the results of previous studies
- could design a diagram to present the effects of oxidative stress on fetuses.
- the manuscript needs extensive revision for language and grammar, as well as in the design of the text.
Minor comments
§ L11: Especially in late pregnancy..
§ L14: .. nutrients to the fetus..
§ L15: the changes in energy
§ L21-25: In the modern pig industry, maintaining the sow performance has become important for achieving the best production targets. Piglets weaned per sow per year (PSY) is an important factor to measure the efficiency of pig farms and the reproductive performance of sows[1]. Genetic selection and early weaning have resulted in the PSY of hyper-prolific sows being able to more than 31[2], which may cause a more metabolic burden for sows and lead to reduced service life. kg
§ L34: The placenta is the only site for the contact
§ L39: stress-induced placental dysfunction…
§ L54: .. as a role in the maintenance of CL[17], a study
§ L63: .. and is also important
§ L68: .. for the establishment
§ L84: .. had a higher number of live-born
§ L92: A study has shown..
§ L93: .. with an extra 2 kg/d of feed
§ L97: .. in the second trimester
§ L106:.. levels decrease in late pregnancy
§ L116: .. the development of the mammary gland
§ L139: A study has shown that high-back fat sows had
Author Response
Thanks for your valuable comments, the modifications to your comments are as follows:
Major comments:
Pointed 1: the text of this manuscript should contain one -two tables, summarizing the results of previous studies.
Response 1: According to your comments, an additional table has been added in the new manuscript.
Pointed 2: could design a diagram to present the effects of oxidative stress on fetuses.
Response 2: According to your comments, an additional diagram about the effect of oxidative stress on fetus has been added in the new manuscript.
Pointed 3: the manuscript needs extensive revision for language and grammar, as well as in the design of the text.
Response 3: We have carefully revised the grammar and English in the new manuscript.
Minor comments: the grammar problem has been revised accordingly in the new manuscript.

Reviewer 2 Report
The authors need to have someone proficient in English edit the manuscript for the grammar. The information is comprehensive but and improved grammar and English is needed to meet the journals high status. I have started to suggest corrections but after the introduction feel it needs an extensive revision
Introduction
Ln 24: remove ‘PSY of hyper-prolific sow being able to more than 31[2], which may cause more metabolic burden for sows, and lead to the reduction of their service life.’
Add ‘PSY of hyper-prolific sows to be than 31[2], and this may cause greater metabolic burden for sows, and potentially to a reduced service life.’ Also what is meant by service life? Is this return to service interval.?
Ln 26: remove ‘At the same time, it was found that the more litter size, the lower birth weight of piglets, and the higher is the variation in piglet birth-weight within the litter[3,4], the increase of variation in birth weights of piglets within litters increases management costs in modern all-in-all-out swine production systems[5].
Add ‘At the same time, the higher litter size, lower birth weight, and the greater variation in piglet birth-weight within the litter[3,4], increases management costs in modern all-in-all-out swine production systems[5].
Ln 31: remove ‘Maternal metabolic status during different stage of pregnancy changes greatly, such as lipid[7] and glucose metabolism[8], which facilitate nutrient supply to the developing fetus and have great impact on reproductive performance’
add ‘Maternal metabolic status during different stages of pregnancy with considerable changes in lipid[7] and glucose metabolism[8], which facilitate nutrient supply to the developing fetus and have a significant impact on reproductive performance’
Ln 33. Remove ‘Maternal metabolism during early gestation is largely anabolic[9], and is often under catabolic conditions in late pregnancy, metabolic disorders at any stage in pregnancy may impair the growth of the fetus and the neonate as well as increase their morbidity and mortality[10]’
Add ‘Maternal metabolism during early gestation is largely anabolic[9], with catabolic conditions prevalent in late pregnancy. Metabolic disorders at any stage in pregnancy may impair the growth of the fetus and the neonate as well as increase their morbidity and mortality[10]’
Ln 36: remove ‘ Placenta is the only site for contact between the fetus and the maternal, which maintains pregnancy and embryonic development and growth by transporting nutrients, participating in the maternal immune response and secreting hormones[11,12].
Add ‘The placenta acta as the essential contact between the separate fetal and maternal circulations, and maintains pregnancy and embryonic development and growth by supporting
transfer of nutrients, immunoglobulins and waste and acts as an endocrine organ [11,12].’
Ln 41: remove ‘the changes of energy and metabolism during pregnancy’
add ‘the changes of energy supply and metabolism during pregnancy’
Author Response
Response to Reviewer 2 Comments
Thanks for your valuable comments, the modifications to your comments are as follows:
The listed suggestion of corrections has been revised in the new manuscript.
Pointed 1: The authors need to have someone proficient in English edit the manuscript for the grammar. The information is comprehensive but and improved grammar and English is needed to meet the journals high status. I have started to suggest corrections but after the introduction feel it needs an extensive revision.
Response 1: We have carefully revised the grammar and English in the new manuscript, and asked a native speaker in this field to edit our manuscript.
Pointed 2: what is meant by service life? Is this return to service interval.?
Response 2: Service life is productive life, which has been revised in the new manuscript.

Round 2
Reviewer 1 Report
I accept the revised manuscript based on my previous comments. I suggest that this revised article is suitable for publication in the current formReviewer 2 Report
A couple of grammar corrections are suggested
Ln 89: ‘sow gilts’ use ‘is it sows or gilts or both and if so ‘sow and gilts’
Ln 111: ‘mmol/Lin’ include space ‘mmol/L in’
Ln 131: remove ‘thickened’ add ‘increased’
Ln 155: remove ‘indicators’ add ‘indicator’
Ln 266: remove ‘breastfeeding’ add ‘lactation’
Ln 283: remove ‘more just’ add ‘more than just’
Ln 330: remove ‘to changes’ add ‘with changes’
In graphic abstract ‘spelling of estorge’?